# Particle Morphology and Elemental Analysis of Lung Tissue from Post-9/11 Military Personnel with Biopsy-Proven Lung Disease

**DOI:** 10.3390/ijerph21010091

**Published:** 2024-01-12

**Authors:** Heather Lowers, Lauren Zell-Baran, Zikri Arslan, Camille M. Moore, Cecile Rose

**Affiliations:** 1U.S. Geological Survey, Geology Geophysics Geochemistry Science Center, Denver, CO 80225, USA; hlowers@usgs.gov (H.L.); zarslan@usgs.gov (Z.A.); 2National Jewish Health, Department of Medicine, Division of Environmental and Occupational Health Sciences, Denver, CO 80206, USA; zellbaranl@njhealth.org; 3Department of Epidemiology, University of Colorado Anschutz Medical Campus, Aurora, CO 80045, USA; 4National Jewish Health, Department of Immunology and Genomic Medicine, Center for Genes, Environment and Health, Denver, CO 80206, USA; mooreca@njhealth.org; 5Department of Biostatistics and Informatics, University of Colorado Anschutz Medical Campus, Aurora, CO 80045, USA; 6Department of Medicine and Environmental and Occupational Health, University of Colorado Anschutz Medical Campus, Aurora, CO 80045, USA

**Keywords:** dust, silica, particulate matter, respirable

## Abstract

The relationship between exposure to inhaled inorganic particulate matter and risk for deployment-related lung disease in military personnel is unclear due in part to difficulties characterizing individual exposure to airborne hazards. We evaluated the association between self-reported deployment exposures and particulate matter (PM) contained in lung tissue from previously deployed personnel with lung disease (“deployers”). The PM in deployer tissues was compared to normal lung tissue PM using the analytical results of scanning electron microscopy and inductively coupled plasma mass spectrometry. The majority of PM phases for both the deployers and the controls were sub-micrometer in size and were compositionally classified as aluminum and zirconium oxides, carbonaceous particles, iron oxides, titanium oxides, silica, other silicates, and other metals. The proportion of silica and other silicates was significantly higher in the retained dust from military veterans with biopsy-confirmed deployment-related lung disease compared to the control subjects. Within the deployer population, those who had combat jobs had a higher total PM burden, though the difference was not statistically significant. These findings have important implications for understanding the role of inhaled inorganic dusts in the risk for lung injury in previously deployed military veterans.

## 1. Introduction

There is increasing recognition of the adverse respiratory health outcomes that can occur following military deployment to Southwest Asia (including Afghanistan and the Horn of Africa) [1,2,3,4,5]. In August 2022, Congress passed The Sergeant First Class Heath Robinson Honoring our Promise to Address Comprehensive Toxins (PACT) Act for military veterans with respiratory illnesses following Vietnam, the Gulf War, and post-9/11era deployments. Among the 23 diagnoses presumptively linked to these military missions, the PACT Act recognizes 11 specific deployment-related respiratory diseases including rhinitis, sinusitis, asthma, bronchiolitis, granulomatous inflammation, and other distal lung abnormalities [6]. The PACT Act implicitly acknowledges the challenges inherent to identifying deployment-specific causal exposures. These challenges include the sheer number of potential inhalational hazards in multiple regions of conflict, the lack of real time exposure monitoring, and the marked variability in individual airborne exposures based on timing, location, duration, intensity, and job duties during military deployment.

Military operations generate airborne hazards, not just at work but also during leisure activities and sleep, due to the environmental realities of many deployment locations. According to the National Academies of Sciences, Engineering, and Medicine, such regional environmental exposures may include air pollution from local point and area sources such as traffic, waste management, and industry [2]. Military operations are frequent contributors, with things such as the exhaust from vehicles, aircraft, and heaters as well as smoke from structural fires, explosive blasts, and burn pits (large areas where tons of waste products including trash, plastics, wood, metal, paints, solvents, munitions, and medical and human waste are burned in the open air with jet fuel added as an accelerant). Some military personnel have job duties that expose them to potentially hazardous airborne vapors, dusts, gases, or fumes. There have been a number of studies focused on characterizing airborne particulate matter (PM) levels in Southwest Asia including a limited number examining in situ lung PM [7,8,9,10,11,12,13]. While these studies vary in the methods used, timing, and location, most have found that PM exposures often exceeded the Environmental Protection Agency’s National Ambient Air Quality Standards (NAAQS) or the World Health Organization’s air quality guidelines for PM2.5 and PM10. Studies also suggest a substantial degree of variability in the exposures experienced by individuals during military theater operations. Information on occupational exposures from specific job duties is particularly limited.

Given the challenges of characterizing individual deployment airborne hazards, we explored whether the analysis of retained lung PM would provide insight into relevant inorganic dust exposures. We characterized the particle morphology and quantified the elemental compositions of PM in lung tissue from symptomatic veterans with biopsy-proven lung disease compared to healthy control lung tissue samples. We also analyzed whether estimated self-reported deployment exposures were associated with the composition of PM retained in lung tissue.

## 2. Materials and Methods

### 2.1. Study Populations

Among 248 patients seen at the Center of Excellence for Deployment-Related Lung Disease at National Jewish Health (NJH), we obtained diagnostic lung biopsies in 65 patients as part of comprehensive clinical evaluations for persistent, unexplained exertional dyspnea and other respiratory symptoms following military deployment to Southwest Asia and Afghanistan between 2001 and 2017 [5]. We compared findings from 24 previously deployed patients with biopsy-confirmed distal lung disease who provided informed consent to participate in our research with 11 control samples from lungs donated to NJH by accident victims, scored by a pulmonary pathologist blinded to the samples’ status [5]. For the control samples, information on smoking status and medical history was provided by the next of kin, and none of the donors were known to have lung disease. We selected control samples from subjects who were as similar as possible in age and smoking status to the deployed study participants.

All the previously deployed study subjects were characterized by reported sex, age, race/ethnicity, smoking status, and smoking pack-years at the time of lung tissue acquisition. All of them had histologic findings of deployment-related distal lung disease (including bronchiolitis, emphysema, peribronchiolar metaplasia, granulomatous pneumonitis, and/or lymphocytic interstitial inflammation) [5]. All of them were administered a detailed questionnaire that elicited information on military deployment-specific exposures including timing and location as well as estimates of proximity, frequency, and intensity of exposure to PM generated from burn pits, sandstorms, and diesel exhaust. We calculated individual weighted respiratory hazards scores using a modified inhalational exposure matrix [14] based on the following formula:(Months Deployed × Frequency of Exposure to Burn Pits in days/month)+ (Months Deployed × Frequency of Exposure to Sandstorms in days/month)+ (Months Deployed × Frequency of Exposure to Diesel Exhaust in days/month)(1)

Additionally, we defined high intensity exposures to burn pit smoke, sandstorms, and diesel exhaust as those occurring once a week or more and low exposures as those reportedly occurring less than weekly. Military occupational specialty codes (MOS) were recorded for each deployment and classified as combat or non-combat [14], as evidence shows that combat jobs have been significantly associated with more frequent exposure to inhalational hazards and more time outdoors than non-combat MOS.

### 2.2. Preparation of Tissue Samples for Analysis

Microtome-cut 50-micron-thick lung tissue scrolls were obtained from formalin-fixed, paraffin-embedded surgical lung tissue blocks. Upper lobe samples were available from all 24 cases, lower lobe samples from 22, and middle lobe samples from three deployers. Information on specific lobes from the control samples was not available. The deployer tissue scrolls were generally larger than the control scrolls (Appendix A).

The tissue mass in each scroll was low, ranging between 2 and 3 mg. The tissue scrolls were treated, and the elemental and particulate concentrations were determined based on tissue volume. The tissue volume was determined by multiplying the area of tissue calculated from scanned slide images by the scroll thickness. The tissue scrolls were first treated with xylene in Teflon tubes (4 mL, Savillex) to remove the paraffin coating. After removing the xylene layer, the tissue samples were digested in hydrogen peroxide (H_2_O_2)_ on a hot block (DigiPrep Cube—any use of trade, firm, or product names is for descriptive purposes only and does not imply endorsement by the U.S. Government—SCP Science) at 90 °C until all the tissue was completely dissolved. The digestates were cooled to room temperature, then sonicated for 30 min in an ultrasonic bath. A volume of 400 µL suspension was funneled through a Millipore filtration setup onto a 25 mm diameter, 0.1 µm pore size track-etched polycarbonate filter. The filter was placed on an SEM sample stub, trimmed to fit, and coated with ~15 nm of carbon for conductivity. The remaining sample was submitted for elemental analysis by inductively coupled plasma mass spectrometry (ICP-MS). It is worth highlighting that our novel methodology for preparing small volumes of tissue with H_2_O_2_ in one tube for all steps, including paraffin removal, tissue digestion, sonication, and centrifugation, avoided multiple filtration steps in order to minimize contamination and sample loss. In addition, H_2_O_2_ is less corrosive to PM than sodium hypochlorite, which is traditionally used. High-purity H_2_O_2_ solutions are also available commercially and have the added advantage of minimizing trace metal contamination to achieve lower detection limits in ICP-MS analyses of small tissue samples. Process blanks were prepared alongside the samples to evaluate possible contamination sources.

### 2.3. Elemental Analysis via ICP-MS

The remaining H_2_O_2_ digestate sample was centrifuged at 13,000 rpm for 30 min on a microcentrifuge (Sorvall Legend Micro 21, Thermo Scientific, Dreieich, Germany) to settle the particulate fraction from the hydrogen peroxide-soluble fraction. After separation, the fractions were digested, and the resulting sample solutions were analyzed via ICP-MS using a Perkin Elmer NexION 2000B ICP-MS instrument equipped with an HF-compatible sample introduction system for a suite of major and minor elements, trace transition metals, and rare earth elements. Due to the small sample sizes, the concentrations for several elements—lithium (Li), magnesium (Mg) calcium (Ca), potassium (K), vanadium (V), selenium (Se), arsenic (As), molybdenum (Mo), strontium (Sr), silver (Ag), barium (Ba), cadmium (Cd), antimony (Sb), cerium (Ce), and Lanthanum (La)—were either consistently below the limits of detection or did not meet the quality assurance/quality control (QA/QC) requirements and were removed from further analysis. Of the remaining elements, silicon (Si), aluminum (Al), manganese (Mn), and titanium (Ti) were considered to represent a geogenic dust “fingerprint”, whereas cobalt (Co), chromium (Cr), copper (Cu), nickel (Ni), lead (Pb), tin (Sn), and uranium (U) were attributed to an anthropogenic fingerprint. The remaining elements, iron (Fe) and zinc (Zn), could reflect both geogenic and anthropogenic sources as well as endogenous tissue sources. Where appropriate, statistical analyses were performed using method-blank-corrected values. The blanks for the digestion methods were prepared and analyzed along with the samples to correct for possible reagent contamination. Additional procedural details and instrumental conditions for the elemental analysis are provided in the Appendix A.

### 2.4. Scanning Electron Microscopy Imaging

An automated particle analysis was performed on an FEI field emission scanning electron microscope operated at 15 kV, a spot size of 5, a working distance of 11 mm, the objective aperture set to 30 mm, and the magnification set to 3000 for a 48 µm wide field of view. The data were collected using the Feature module of the Oxford Instruments Aztec^®^ Micoanalysis Software suite (v 6.0). The particle selection thresholds were set to ignore the filter and select all the pixels with a higher backscattered electron intensity than the filter (Figure 1). The stopping criteria were set to 56 fields of view (0.06% of the filter area) or 1500 features, whichever occurred first. Energy dispersive spectra (EDS) were acquired for 5 s per feature with an Oxford Instruments x-Max 50 mm^2^ silicon drift detector. In addition to the qualitative chemical analysis of the particles, the area, aspect ratio, breadth (shortest linear distance of ferret diameter), length (longest linear dimension of ferret diameter), and perimeter were recorded [15]. The particles were initially classified into 65 phases based on their qualitative chemical composition and our interpretation of the EDS spectra.

The phases were then combined into eight compositional categories based on mineral groups and possible exposure sources. The AlZr category contains aluminum oxide-, zirconium oxide-, and zirconium-rich particles that may reflect occupational exposure to abrasive dusts and/or welding/casting fumes [16,17,18,19,20]. The C category contains carbonaceous particles, which may reflect incompletely digested tissue, diesel particulate, or inorganic carbon particulate. The analysis conditions used were not conducive to differentiating the phases within the C category. The Fe category contains endogenous iron and iron phosphate phases as well as iron oxide, some of which may be linked to geogenic and anthropogenic sources. The iron oxide phases containing variable amounts of titanium were placed in the GeoFe category, as they likely represent geogenic sources. The titanium and titanium oxide phases were placed in the Ti category. The titanium phases are ubiquitous in the environment from geogenic sources as well as from anthropogenic sources in food, cosmetic, drug, and other industrial uses. Silica (Si) was analyzed separately from other silicate minerals including micas, clays, and feldspars (OSi) due to its known toxicity, causing silicosis. Finally, other metals that are not components of the AlZr, GeoFe, Fe, or Ti categories, such as the chromium, copper, nickel, and zinc phases, were placed in the O category.

### 2.5. Statistical Analysis

The demographic and smoking characteristics for the subjects were compared using *t*-tests for the continuous variables and Fisher’s Exact Tests for the categorical variables. To account for multiple measurements within each sample and multiple samples from some subjects, the morphology data were log-transformed and compared between the deployers and the controls using a mixed model (using SAS software’s PROC MIXED) with random effects for each sample and each subject.

For one lobe each in two subjects, sequential cuts were analyzed as repeated samples to access variability within the lung tissue block. The counts for each sample by PM classification are presented in Appendix A. These results as well as the ICP-MS analysis are visually displayed side-by-side in Appendix A. To assess method variability, three samples with differing PM concentrations were re-analyzed for a total of three runs each (Appendix A, Appendix A).

We compared the composition of PM by category between the previously deployed and control samples using a negative binomial mixed model (using SAS software’s PROC GLIMMIX) to accommodate for the over-dispersion of the data and used a random effect for each subject to account for repeated measures. The PM classification count data without blank correction was used. The model included an offset for the log of the total particle counts for each sample. Using the same methodology, PM composition comparisons between the upper and lower lobes were made among 20 of the 24 deployers for whom both upper and lower lobe samples were available. All the models were adjusted for age.

We compared the elemental abundance data between the deployers and the controls using a maximum likelihood approach to account for the censoring of values above and below the lower and upper limits of detection, using SAS software’s PROC NLMIXED. The models included a random effect for each subject to account for repeated measures. The concentrations from the total fraction were corrected for the digestion method blanks and then log-transformed to meet the normality assumption. The tissue volume and age were included as covariates to account for differences in the initial sample size and the age of the subjects. When more than half of the concentrations were below the lower limit of detection, we compared groups using a logistic regression model (using SAS software’s PROC GLIMMIX) with a random effect for each subject and a binary outcome for detected/non-detected concentrations, again adjusting for differences in tissue volume and age.

We compared specific PM classifications between the deployers with continuous weighted respiratory hazards exposure scores. We used negative binomial mixed models (using PROC GLIMMIX) with a random effect for each subject and adjusted all the models for age. An offset was applied using the log of the total particle count for each sample where appropriate, or we included the log of the tissue volume and the log of the fraction of the filter examined as covariates in these models. We further adjusted the comparisons of the carbonaceous PM fraction for the reported diesel exhaust exposures.

All the primary analyses were performed in SASv.9.4 with figures created in Rv.1.3.1093.

## 3. Results

The demographic and deployment characteristics are summarized in Table 1. Despite efforts to match the controls on sex, age, and smoking status, the deployers were more likely to be male, younger, and less likely to smoke. Smoking information was missing for five of the eleven controls, limiting our ability to adjust for this potential confounder in most analyses.

The deployed subjects had a median of three unique military deployments and a median total deployment duration of approximately 2.4 years. Nine (37.5%) had been deployed to Iraq, six (25.0%) to Afghanistan, and nine (37.5%) to both locations. In addition to Iraq and Afghanistan, nine had been deployed to other Southwest Asian locations (mainly Qatar and Kuwait). Most of them reported high-intensity exposures to burn pit smoke (22/24, 91.7%) and diesel exhaust (22/22, 100.0%), while approximately one-quarter of them reported high-intensity exposure to sandstorms (6/23, 26.1%). The majority (8/13, 61.5%) worked in a combat military occupational specialty code.

We compared the composition of lung tissue findings between the deployers and the controls based on the eight broad PM categories (Table 2 and Figure 2). Notably, the proportion of silica and other silicates was significantly higher in the deployer samples compared to the controls. While the proportion of titanium phases was also higher in the deployer samples, this finding was not statistically significant after correcting for multiple comparisons. The proportion of iron oxides (Fe) was also higher in the deployer samples but was not statistically significant. The control samples had higher proportions of carbonaceous materials, aluminum and zirconium oxides, geogenic iron phases, and other metals (O), but, for all of them, the comparisons were not statistically significant. The magnitude of the differences in composition are presented in Appendix A. The proportion of silica was 359% (95% CI: 70%, 1139%) higher in the deployers compared to the controls, and the proportion of other silicates was 236% (95% CI: 58%, 613%) higher.

Replicate measurements (Appendix A) of the same filters showed that the variability was greatest for AlZr. Among the 20 deployers with both upper and lower lobe lung tissue samples having been analyzed, the proportion of each of the PM categories was similarly abundant in both lobes (Table 3). Only three middle lobe samples were available, limiting comparisons.

The particles were mostly equidimensional, and fewer than 1000 of the 79,280 particles had aspect ratios > 3 ([15], Figure 1). Since the particle morphology was mostly equidimensional, only the length and area findings are given in Table 4 and in Appendix A. The median particle length (µm) and area (µm^2^) were generally smaller in the deployer tissue samples compared to the controls but not significantly different. The size distribution of the particles in the upper and lower lobes was not different (Table 5).

The elemental analysis of the acid-digested PM and H_2_O_2_ fractions showed no differences in concentrations between the deployer and control samples (refer to Figure 3 and Table 6). The content of each element is greatest in the particle fraction, suggesting that the elements are bound in mineral or other solid phases and not to the tissue. Exceptions are Zn and Cu, both known to be endogenous elements. For some of the subjects, the concentrations of several elements (Mn, Ni, Cr, Pb, Sn) were relatively high in the H_2_O_2_ fraction. These dissolved elements may have originated from the tissue or via partial dissolution of reactive particles.

We explored whether the self-reported exposures to known inhalational hazards during deployment were linked to the PM measured in the lung tissue samples. The exposure scores were not significantly associated with PM proportion or the total PM burden (Appendix A). The deployers with a combat-related MOS had a 36% higher total PM burden (95% CI: 10% lower to 103% higher) than those without a combat job, but this finding was not statistically significant (*p* = 0.13).

## 4. Discussion

We analyzed PM from lung tissue and, for the first time, compared morphology and elemental composition in previously deployed military personnel with biopsy-confirmed deployment-related lung disease to controls. The majority of PM phases in both the deployer and control tissue samples were sub-micrometer in size and were categorized as silica, other silicate minerals, iron oxides, titanium oxides, aluminum and zirconium oxides, other metals, and carbonaceous particles. We found that the proportion of silica and other silicates was significantly higher in the lung tissue samples from previously deployed military veterans with lung disease compared to the controls. Self-reported exposures were not significantly associated with PM concentrations, but the total PM burden was elevated (though not statistically significantly) among the participants with a combat MOS.

Comparing the mean PM lengths, we found no significant differences between the deployers and the controls, though the Ti category of particles was generally smaller in the deployer samples. Previous studies have shown that the fraction of respirable (<2.5 micron) dust from Camp Victory in Iraq contains Ti and other silicate minerals with sharp, angular edges, findings which linked with lung inflammation in mice exposed to the dust [21]. “Hot spots” of titanium and iron were found in the lung tissue of a soldier who developed interstitial and peribronchiolar fibrosis following deployment to Iraq and Kuwait [22]. We found no significant difference in the proportion of Ti contained in the lung tissue of the deployers compared to the controls.

Not surprisingly, the mean particle lengths in all the tissue samples analyzed were smaller than the mean diameter of aerosolized dust obtained from a point source in Bagram, Afghanistan (0.829 µm), that was used in several animal experiments [15,23,24]. The retained PM was also smaller than most of the particulate matter collected on air filters from multiple locations throughout Southwest Asia, where less than 2% (by mass) were between 0.5 and 1.0 µm in size [8]. In our study, the PM recovered from the lung lacked the clay or iron oxide coatings observed by Engelbrecht et al. [8,25] and Lowers et al. [15] in bulk dust samples, likely due to loss during particle processing. The differences in size between lung PM and Southwest Asia dust suggest that smaller particles bypass filtration and mucus clearance processes in the nose and proximal airways and are translocated deeper into the lung, where lung host defenses act to mitigate toxicity [26]. Similarly to other reports, we did not see differences in the particle load, type, or size among different lung lobes [26,27].

A recent study of lung biopsy samples from deployed and non-deployed military personnel used automated field emission scanning electron microscopy with energy-dispersive X-ray spectroscopy to characterize inorganic in situ particulate matter and found an abundance of clays, feldspars, silica, titanium dioxides, and metals [12]. The in situ PM was typically found in clusters, which made the interpretation of the EDS spectrum difficult and limited particle size analysis. The method we used to dissolve the tissue and disperse PM on the filter minimized the problem with overlapping/clustered particles and enabled measurements of particle morphology. Despite using different approaches, both studies identified many of the same PM categories. However, the proportions of silica, silicates, and other elemental components were not analyzed or compared in the Hayden et al. [12] study. A recent study of lung tissue from patients in a non-occupational setting examined the distribution and structural fingerprint of metals in pulmonary PM [27]. Those with the highest lung tissue retention were Al, Cd, Cr, Ba, Ni, Ti, Sn, V, and Sb, while Ca, Mg, and Zn had the lowest retention. We observed similarly elevated concentrations of Al, Cr, Ni, Ti, and Sn in the deployer samples compared to the controls but found higher concentrations of Zn. The concentrations of Cd, Ba, V, Ca, and Sb in our study were either consistently below limits of detection or did not meet the QA/QC requirements for analysis.

Our bulk composition analyses demonstrated that most of the metals were retained in the PM fraction. However, a small concentration of most elements was detectable in the H_2_O_2_ fraction, with Cr, Sn, and Pb being highest in concentration. This suggests at least two forms of binding for each of these metals. As, Pb, Ni, and Sb were elevated compared to their crustal abundance in the point source dust [15] and air filter samples collected around southwest Asia [8,25,28]. The antimony (Sb) concentrations were below the limit of detection in all the samples in this study and were not evaluated. The exposure sources of these metals in military deployment locations are likely anthropogenic, e.g., from smelters, burn pits, and leaded gasoline. However, we were unable to specify the sources of Pb in the subject samples to determine its potential metal bioavailability. Cr, Ni, and Sn occur as metal oxides and in combination with other metals such as Fe, Mn, and Cu and likely reflect anthropogenic sources as well. The silicon (Si) in the PM fraction from silica and other silicate minerals is likely of geogenic origin. We attributed the H_2_O_2_ soluble Si to reactive silicate particles including water-soluble metasilicate salts. The total concentration of elements present was similar in the controls and the deployers. Lacking the occupational and environmental exposure histories of the control subjects and the deployers (beyond deployment), we have little insight into the sources of some of the element contents identified.

Our study had several strengths. First, we had access to lung tissue samples from both previously deployed military veterans and from controls and were able to make comparisons regarding PM proportions not previously reported in the published literature. Second, detailed information was available regarding hazardous exposures for cases during military deployments, and smoking histories were available for the majority of the study samples. Third, we used sophisticated analytical techniques with extremely sensitive detection limits to characterize the retained lung PM and elements from a wide spectrum of anthropogenic, geogenic, and mixed exposure sources. Fourth, the potential contamination during sample handling and preparation was accounted for by the analysis of process blanks along with the samples.

Our study also had several limitations. First, the small numbers of both deployer and control samples limited our ability to detect differences between the groups. Despite the limited sample numbers, the finding of significantly higher proportions of silica and silicates in the veterans with deployment-related lung disease was noteworthy. Second, the control group tended to be older and smoked more than the deployer group, biasing the results towards the null. We may have found other significant differences had more non-smoking controls been included. Third, complete occupational and environmental exposure histories for both the controls and the military deployers were unknown, potentially confounding our results. Additionally, details for the controls were provided by the next of kin, potentially resulting in information bias, though we would not anticipate that this would have been systematic in either direction which would bias the results upwards or downwards. Fourth, factors that may lead to increased vulnerability to airborne hazards such as exposure to temperature extremes, stress, noise, and acute infections—common in military deployment locations—could not be analyzed.

Several instrumentation and laboratory analytical challenges further contributed to this study’s limitations. First, the SEM and ICP-MS results reflect only the retained inorganic PM, and contributions to lung injury from non-retained hazardous inhalants such as microbial bioaerosols, polyaromatic hydrocarbons, volatile organic compounds, pesticides, and allergens could not be analyzed. Second, only small amounts of material (2–3 mg) were available for use in two separate analytical techniques. These small volumes resulted in analyses that were near the detection limit for the ICP-MS analysis, so comparisons between the deployers and the controls were not possible for many elements. Third, the replicate analysis of the samples for SEM showed some variability, especially for the AlZr category.

## 5. Conclusions

Using sophisticated analytical techniques, we found that the proportion of silica and other silicates was significantly higher in the lung tissue samples from previously deployed military veterans with biopsy-confirmed lung disease compared to the controls. Our findings suggest that deployment-related lung injury may be causally linked to inorganic dust exposure in military theaters, with important implications for exposure control in these occupational settings. Our analytical approach using lung tissue digestion, automated particle analysis, and comparison of elemental proportions in lung tissue from different populations adds to the growing literature focused on understanding the role of inhaled PM in respiratory diseases.

## Figures and Tables

**Figure 1 ijerph-21-00091-f001:**
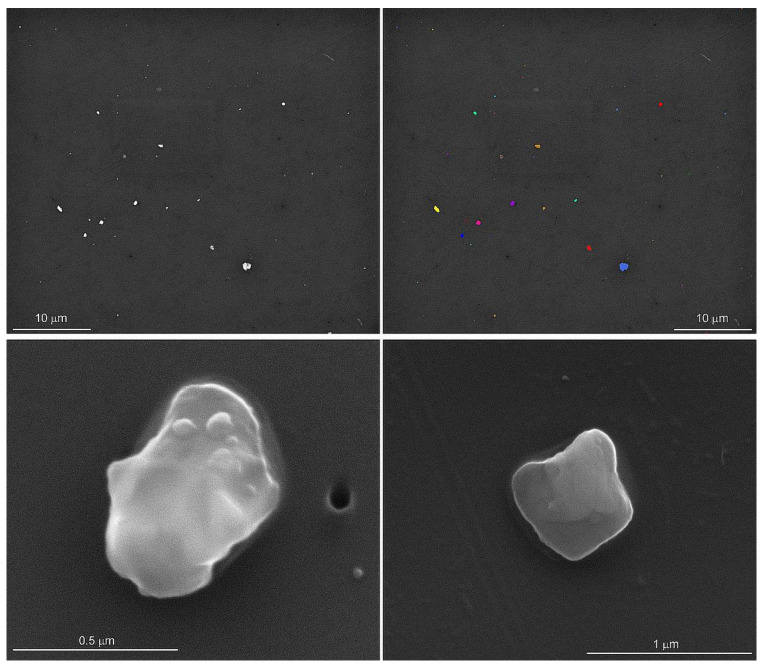
Backscattered scanning electron microscope image of particles dispersed on the filter (**upper left**). The thresholds were set to separate the particles from the filter for the automated particle analysis (**upper right**). The secondary electron images of silica (**lower left**) and potassium aluminosilicate (**lower right**) show rounded edges and sizes typically less than one micrometer in length. Particle morphology and elemental analyses were conducted on lung tissues from control subjects and on lung tissues from previously deployed military personnel with lung disease.

**Figure 2 ijerph-21-00091-f002:**
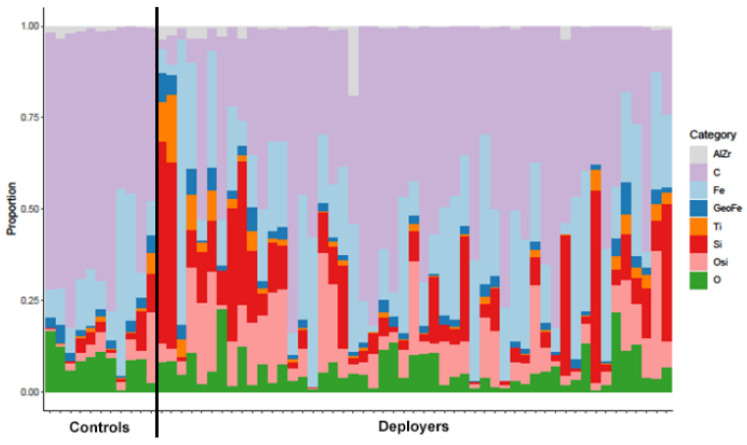
Particulate matter proportions of aluminum and zirconium oxides (AlZr), carbonaceous phases (C), iron (Fe), geogenic iron (GeoFe), titanium phases (Ti), silica (Si), other silicates (OSi), and other metals (O) in tissue samples from the lungs of previously deployed military personnel with lung disease (deployers) and controls (separated by solid black line). Each bar represents a unique sample.

**Figure 3 ijerph-21-00091-f003:**
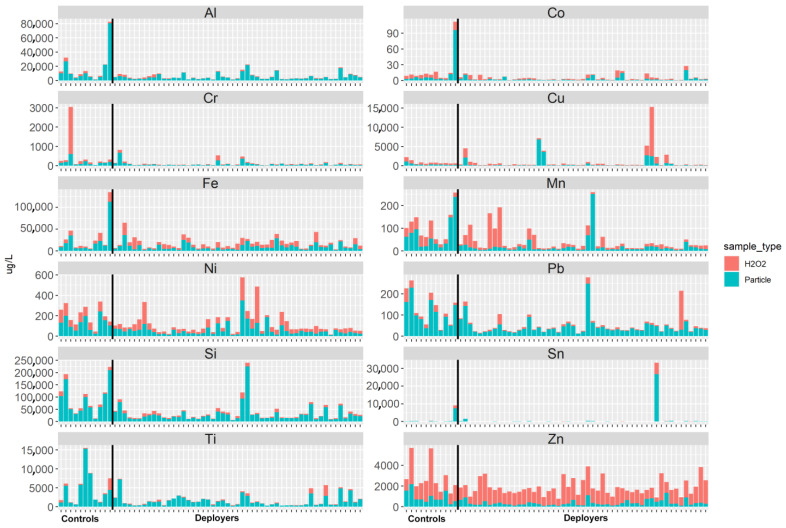
Selected element abundance in the tissue as represented by the H_2_O_2_ fraction (red) and in the particulate matter (blue) fraction from the tissue slices of lung samples from previously deployed military personnel (deployers) and controls.

**Table 1 ijerph-21-00091-t001:** Demographic and deployment characteristics of previously deployed military personnel with lung disease (deployers) and control subjects.

Variable	Statistic	Deployers (n = 24) ^1^	Controls (n = 11) ^1^	*p*-Value ^2^
**Demographic Characteristics**				
Male	n (%)	21 (87.5)	5 (45.5)	0.01
Age (years)	mean ± SD	41.0 ± 7.8	54.8 ± 18.4	0.03
Smoking status	n (%)			**0.002**
Current		1 (4.2)	1 (9.1)	
Former		8 (33.3)	2 (18.2)	
Never		15 (62.5)	3 (27.3)	
Missing		0	5 (45.5)	
Smoking pack-years	mean ± SD	5.0 ± 4.5	12.2 ± 19.8	0.59
Non-Hispanic White	n (%)	22 (91.7)	8/10 (80.0)	0.24
**Deployment Exposure Characteristics**				
Number of deployments	median (range)	3 (1–8)	--	--
Total months deployed	mean ± SD	29.1 ± 20.5	--	--
Deployment location	n (%)			
Iraq		9 (37.5)	--	--
Afghanistan		6 (25.0)	--	--
Both		9 (37.5)	--	--
Reported high-intensity exposure to:	n (%)		--	--
Burn pit smoke		22/24 (91.7)	--	--
Sandstorms		6/23 (26.1)	--	--
Diesel exhaust		22/22 (100)	--	--
Exposure index score:	median (range)			
Burn pit smoke		631 (0–2760)	--	--
Sandstorms		46 (0–551)	--	--
Diesel exhaust		587 (60–2434)	--	--
Cumulative		1581 (255–5349)	--	--
Combat MOS ^3^	n (%)	8/13 (61.5)	--	--

^1^ Differences in denominator reflect missing data. ^2^ Where appropriate, continuous variables were compared via *t*-test, and categorical variables were compared via Fisher’s Exact Test. *p*-values < 0.01 are bold (adjusting for multiple comparisons with a Bonferroni correction for five tests). ^3^ MOS = military occupational specialty code.

**Table 2 ijerph-21-00091-t002:** Summary of composition (%) of particulate matter (PM) in tissue samples from the lungs of previously deployed military personnel with lung disease (deployers) and controls by category, median (range).

Category	Components ^1^	Deployer Samples (n = 51) ^2^	Control Samples (n = 11)	*p*-Value ^3^
AlZr	Aluminum oxide, zirconium oxide	0.4 (0–19.3)	1.4 (0.2–3.4)	0.25
Carbonaceous (C)	Carbonaceous particles	45.8 (2.5–83.7)	67.8 (44.1–80.6)	0.03
Fe	Iron oxide, iron phosphate	20.0 (0.2–78.0)	9.3 (6.7–51.3)	0.08
GeoFe	Iron titanium oxide	2.1 (0–8.7)	2.4 (0.4–5.1)	0.96
Titanium Oxide (Ti)	Titanium and titanium oxides	1.0 (0–18.3)	0.4 (0–5.6)	0.04
Silica (Si)	Silica	7.3 (0.2–55.2)	1.8 (0–10.8)	**0.004**
Other silicates (OSi)	Aluminosilicates and other silicates	7.7 (0.4–34.9)	2.2 (0.6–19.4)	**0.003**
Metals (O)	Metals not in the other categories: steel, tin, nickel, copper, zinc, etc.	4.5 (0.2–22.4)	8.7 (0.4–16.5)	0.13

^1^ Refer to Lowers et al., 2023, data release [15] for a full accounting of the phases identified. ^2^ Upper lobe samples were available from all 24 cases, lower lobe samples from 22, and middle lobe samples from three deployers. Additionally, sequential cuts for two samples were included, yielding 51 total samples. ^3^ The composition of PM categories was compared between the deployers and the controls using a negative binomial mixed model with a random effect for each subject. The PM classification count without blank correction was used. An offset was applied with the log of the total particle count for each sample. The comparisons were adjusted for age. *p*-values < 0.006 are bold (adjusting for multiple comparisons with a Bonferroni correction for eight tests).

**Table 3 ijerph-21-00091-t003:** Summary of proportion (%) of particulate matter (PM) by category and lung tissue lobe (upper, middle, lower) among previously deployed military personnel with lung disease (deployers), median (range). Aluminum and zirconium oxides (AlZr), carbonaceous phases (C), iron (Fe), geogenic iron (GeoFe), titanium phases (Ti), silica (Si), other silicates (OSi), and other metals (O).

Category	Upper (n = 25)	Middle (n = 3)	Lower (n = 23)	*p*-Value ^1^
AlZr	0.3 (0–3.6)	0.4 (0–1.0)	0.7 (0–19.3)	0.11
C	45.8 (2.9–77.0)	61.5 (23.3–82.6)	42.3 (2.5–83.7)	0.83
Fe	21.7 (0.2–78.0)	6.5 (4.4–20.0)	18.1 (1.6–40.9)	0.30
GeoFe	2.2 (0.2–4.8)	1.3 (0.9–1.8)	2.1 (0–8.7)	0.52
Ti	1.4 (0–5.5)	3.1 (0.1–3.6)	0.4 (0–18.3)	0.52
Si	5.1 (0.8–52.7)	13.8 (1.5–37.7)	9.4 (0.2–55.2)	0.89
OSi	7.7 (0.9–32.7)	7.1 (1.4–10.7)	5.3 (0.4–34.9)	0.51
O	4.5 (0.2–21.5)	6.4 (3.8–6.9)	4.9 (0.5–22.4)	0.39

^1^ The proportion of PM categories was compared between upper and lower lobes using a negative binomial mixed model with a random effect for each subject. PM count data by compositional category without blank correction was used. An adjustment offset was applied using the log of the total particle count for each sample. The comparisons were adjusted for age. Sequential cuts for two samples were included. A total of 20 out of the 24 deployers had both upper and lower lobes available and were included in this analysis.

**Table 4 ijerph-21-00091-t004:** Median (range) length and area of in situ lung tissue particles from the lungs of previously deployed military personnel with lung disease (deployers) and controls. Aluminum and zirconium oxides (AlZr), carbonaceous phases (C), iron (Fe), geogenic iron (GeoFe), titanium phases (Ti), silica (Si), other silicates (OSi), and other metals (O).

Category	Count ^1^		Length (µm)			Area (µm^2^)		
	Deployers	Controls	Deployers	Controls	*p*-Value ^2^	Deployers	Controls	*p*-Value ^2^
Overall	67,023	12,257	0.2 (0.1–16.0)	0.2 (0.1–16.9)	0.46	0.02 (0–84.1)	0.03 (0–44.6)	0.68
AlZr	679	118	0.3 (0.1–8.3)	0.4 (0.1–2.8)	0.06	0.05 (0–27.9)	0.06 (0–3.1)	0.07
C	31,026	7324	0.2 (0.1–15.0)	0.2 (0.1–12.1)	0.54	0.01 (0–76.8)	0.02 (0–44.6)	0.59
Fe	14,734	2406	0.2 (0.1–8.1)	0.3 (0.1–3.7)	0.59	0.02 (0–19.7)	0.03 (0–7.2)	0.68
GeoFe	1472	291	0.3 (0.1–3.9)	0.3 (0.1–2.7)	0.78	0.05 (0–6.3)	0.06 (0–2.6)	0.62
Ti	1109	157	0.3 (0.1–2.8)	0.4 (0.1–2.0)	0.03	0.06 (0–2.4)	0.09 (0–1.0)	0.05
Si	7618	464	0.3 (0.1–16.0)	0.3 (0.1–16.9)	0.26	0.03 (0–84.1)	0.04 (0–38.1)	0.30
OSi	7105	636	0.4 (0.1–10.0)	0.4 (0.1–6.3)	0.29	0.06 (0–41.5)	0.09 (0–19.6)	0.30
O	3280	861	0.2 (0.1–15.9)	0.2 (0.1–9.8)	0.74	0.03 (0–39.9)	0.03 (0–30.2)	0.69

^1^ Counts were not corrected for differences in tissue volume or count area. ^2^ Variables were heavily skewed to smaller values and were log-transformed prior to analysis. The comparisons were made via a mixed model with random effects for each sample and subject in order to account for repeated measures.

**Table 5 ijerph-21-00091-t005:** Summary of size distribution data for in situ lung tissue particles by lobe among the lungs of previously deployed military personnel with lung disease (deployers), median (range).

Parameter	Upper (n = 25)	Middle (n = 3)	Lower (n = 23)	*p*-Value ^1^
Length (µm)	0.2 (0.1–15.8)	0.2 (0.1–15.9)	0.2 (0.1–16.0)	0.72
Area (µm^2^)	0.02 (0–76.8)	0.02 (0–39.9)	0.02 (0–84.1)	0.79

^1^ Variables were heavily left skewed and were log-transformed before analysis. The comparisons were made via a mixed model with random effects for each sample and subject in order to account for repeated measures. Sequential cuts for two samples were included. A total of 20 out of the 24 deployers had both upper and lower lobes available and were included in this analysis.

**Table 6 ijerph-21-00091-t006:** Comparison of selected element abundance in the tissue of lung samples from previously deployed military personnel (deployers) and controls. The concentrations (µg/L) are for tissue volume and reflect the total tissue concentration (particulate matter [PM] + H_2_O_2_ fraction).

Element	Deployer Samples (n = 51)		Control Samples (n = 11)		*p*-Value ^2^
	Detection Frequency	Median (Range) ^1^	Detection Frequency	Median (Range) ^1^	
Al	100%	54.6 (15.1–277)	100%	26.5 (19.6–562)	0.85
Co	90%	0.04 (0.02–0.4)	73%	0.1 (0.02–0.8)	0.33
Cr	100%	0.9 (0.2–4.9)	100%	0.6 (0.4–12.5)	0.40
Cu	100%	3.8 (1.5–98.4)	100%	2.8 (1.6–10.6)	0.45
Fe	100%	197 (28.6–1275)	100%	61.2 (19.0–915)	0.50
Mn	65%	0.4 (0.2–1.9)	64%	0.5 (0.3–1.7)	0.14
Ni	100%	1.0 (0.4–5.0)	100%	0.7 (0.4–1.8)	0.42
Pb	100%	0.5 (0.3–3.9)	100%	0.5 (0.3–1.1)	0.89
Si	100%	314 (164–1792)	100%	222 (198–1509)	0.32
Sn	98%	1.0 (0.5–6.2)	91%	0.6 (0.4–2.6)	0.87
Ti	96%	14.7 (2.5–41.7)	73%	6.6 (3.1–30.7)	0.89
U	20%	0.02 (0.01–0.1)	18%	0.02 (0.02–0.03)	0.49 ^
Zn	100%	21.4 (6–94.7)	100%	11.0 (4.6–23.3)	0.87

^1^ Median and range only include measured values above the limit of detection and below the highest calibration value. ^2^ Elemental abundance data were compared between deployers and controls using a mixed model (using PROC NLMIXED) with a random effect for each subject and likelihood estimates that account for both high and low censoring. The concentrations from the total fraction were log-transformed to meet model assumptions. The models were adjusted for the log of the tissue volume and age. ^ Elemental abundance comparisons for elements with more than half of the concentrations censored were analyzed using a mixed model (using PROC GLIMMIX) with a random effect for each subject and a binary outcome for detected/non-detected concentrations. The models were adjusted for the log of the tissue volume and age.

## Data Availability

The data that support the findings of this study are openly available in Heather A Lowers, Zikri Arslan, William Benzel, David Roth, Terry Plowman, Kate Campbell-Hay, Brian Wong, Karen Mumy, Greg Downey, Cecile Rose, Lauren Zell-Baran, Brian Day, Hong Wei Chu, Max Seibold, and Reena Berman, 2023, Geochemical Analysis of Bulk Dust and Human Respiratory Cells and Fluids in Research on Deployment-related Lung Injury: U.S. Geological Survey Data Release at https://doi.org/10.5066/F7KK98W4 accessed on 5 June 2023.

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
