# Peer review of "Particle Morphology and Elemental Analysis of Lung Tissue from Post-9/11 Military Personnel with Biopsy-Proven Lung Disease"

_ijerph, 2024, doi:10.3390/ijerph21010091_

Round 1

Reviewer 1 Report

Comments and Suggestions for Authors

The study entitled "Particle morphology and elemental analysis of lung tissue from 2 post-9/11 military personnel with biopsy-proven lung disease" measured the exposure to particulate matters (PM) in the lung tissue of military personnel deployers in the southwest Asia and Afghanistan. The study overall is presented well. I only had concerns about the sample size and effect of smoking patterns of the personnel that might have led to the presence of PM in the lung tissue. However, authors have included these points in the limitations of the study and therefore, in my view, no changes are required in the manuscript as such and can be accepted in its present form. 

Author Response

We appreciate this reviewer’s strong support and positive comments on our manuscript.

Reviewer 2 Report

Comments and Suggestions for Authors

The initiative to investigate such a relevant topic implies that the information contained in the references is the most relevant and up-to-date. We recommend reviewing the following articles to support the hypothesis of pulmonary disease due to exposure to particulate matter in the lungs. (https://doi.org/10.1016/j.ecoenv.2022.113324) (https://academic.oup.com/milmed/article/186/Supplement_1/784/6119473).

Although the way of collecting the information is appropriate, I consider that the information that was provided by family members and other people other than the study participants should have presented some quality criteria, in addition to being relevant, I consider it should have been described in the objectives of the study. Therefore, I do not consider it important since they can be taken into account as inclusion/exclusion criteria.

Although it is described by the authors that the method used to dissolve the tissue and disperse the PM allows better medication of the particles. What technological and innovative implications does this research present that will allow the analysis of samples to be approached more accurately? Consider is a factor that should be mentioned in more detail.

Author Response

Reviewer #2:

  1. The initiative to investigate such a relevant topic implies that the information contained in the references is the most relevant and up-to-date. We recommend reviewing the following articles to support the hypothesis of pulmonary disease due to exposure to particulate matter in the lungs. https://doi.org/10.1016/j.ecoenv.2022.113324

Response:  We thank the reviewer for this suggestion and agree that this article provides additional insight into the metals that are more likely to be found in lung tissue. We have added the following sentences to the Discussion on page 12: “A recent study of lung tissue from patients in a non-occupational setting examined the distribution and structural fingerprint of metals in pulmonary PM [32]. Those with the highest lung tissue retention were Al, Cd, Cr, Ba, Ni, Ti, Sn, V, and Sb, while Ca, Mg, and Zn had the lowest retention. We observed similarly elevated concentrations of Al, Cr, Ni, Ti, and Sn in deployer samples compared to controls, but found higher concentrations of Zn. Concentrations of Cd, Ba, V, Ca, and Sb in our study were either consistently below limits of detection or did not meet QA/QC requirements for analysis.”

  1. Although the way of collecting the information is appropriate, I consider that the information that was provided by family members and other people other than the study participants should have presented some quality criteria, in addition to being relevant, I consider it should have been described in the objectives of the study. Therefore, I do not consider it important since they can be taken into account as inclusion/exclusion criteria.

Response: We agree that the information collected about controls from next of kin is a potential study limitation, but felt these samples should still be included since access to adequate lung tissue from well-characterized healthy participants is challenging.  Along with the other limitations of control samples in the Discussion, we added the following sentence on page 12 to further address the reviewer’s point: “Second, the control group tended to be older and smoked more than the deployer group, biasing results towards the null. We may have found other significant differences had more non-smoking controls been included. Third, complete occupational and environmental exposure histories for both controls and military deployers were unknown, leading to potential confounding from unknown exposures. Additionally, details for controls were provided by next of kin, potentially resulting in information bias, though we would not anticipate this would have been systematic in either direction that would bias results high or low.

  1. Although it is described by the authors that the method used to dissolve the tissue and disperse the PM allows better medication of the particles. What technological and innovative implications does this research present that will allow the analysis of samples to be approached more accurately? Consider is a factor that should be mentioned in more detail.

Response: 

We thank the reviewer for the suggestion to better identify the novel methods we employed to more accurately analyze the extremely small tissue volumes we were working with. To address this comment, we have added the following sentences to the Methods on page 3: “To highlight, our novel methodology to prepare small volumes of tissue with H2O2 utilized one tube for all steps including paraffin removal, tissue digestion, sonication, and centrifugation which avoided multiple filtration steps to minimize contamination and sample loss. In addition, H2O2 is less corrosive to PM than sodium hypochlorite which is traditionally used. High-purity H2O2 solution is also available commercially, and has the added advantage of minimizing trace metal contamination to achieve lower detection limits in ICP-MS analysis of small tissue samples.”

Reviewer 3 Report

Comments and Suggestions for Authors

In this article, lung tissue from post-9/11 military soldiers with biopsy-proven lung illness were examined to evaluate particle morphology and elemental analysis. In my opinion, the article needs a careful revision and reworking to be published. Some observations (not exhaustive):

·   The abstract should be restructured according to the IJERPH style: it is essential to include notes on material, methods and results.

· Basically, it is very difficult to correlate exposure (as described) with biological outcomes; furthermore, this is one of the outcomes of this research.

·  However, analytical results related to lung tissue are well represented.

·  Heterogeneous data are reported in the discussion, so no specific conclusion could be reached; in fact, for some metals, the source could not be specified so as to determine potential bioavailability.

· In addition, confusion related to unknown occupational/environmental exposure still remains an underlying bias.

Author Response

Reviewer #3:

In this article, lung tissue from post-9/11 military soldiers with biopsy-proven lung illness were examined to evaluate particle morphology and elemental analysis. In my opinion, the article needs a careful revision and reworking to be published. Some observations (not exhaustive):

  1. The abstract should be restructured according to the IJERPH style: it is essential to include notes on material, methods and results.

Response: The guidelines provided by the journal indicate that the abstract should be a single paragraph limited to 200 words that does not include headings.

The Abstract describes materials and methods as follows: “We evaluated the association between self-reported deployment exposures and particulate matter (PM) contained in lung tissue from previously deployed personnel with lung disease (“deployers”). PM in deployer tissue was compared to normal lung tissue PM using analytical results of scanning electron microscopy and inductively coupled plasma mass spectrometry.”

Abstract results are described as follows: “The majority of PM phases for both deployers and controls are sub-micrometer in size, and were compositionally classified as aluminum and zirconium oxides, carbonaceous particles, iron oxides, titanium oxides, silica, other silicates, and other metals. The proportion of silica and other silicates was significantly higher in retained dust from military veterans with biopsy-confirmed deployment-related lung disease compared to control subjects. Within the deployer population, those who had combat jobs had higher total PM burden, though the difference was not statistically significant.”

The required structure and strict word limitations for the journal do not permit additional explanation.

  1. Basically, it is very difficult to correlate exposure (as described) with biological outcomes; furthermore, this is one of the outcomes of this research.

Response:  We agree with the reviewer that exposure data are especially lacking among deployed military personnel.  As described in the 2020 National Academies of Sciences, Engineering, and Medicine (NAS) report cited in our manuscript [Respiratory Health Effects of Airborne Hazards Exposure in the Southwest Asia Theater of Military Operations, NAS Consensus Study Report], exposure characterization is a pervasive challenge in studies of effects of exposure on military personnel.  Basic information is often lacking on what they were exposed to where and when, at what level, over what time period, and with what frequency.  Our research provides a novel approach that helps address the gaps in knowledge concerning in-theater airborne exposures linked to respiratory health outcomes.

  1. However, analytical results related to lung tissue are well represented.

Response:  We appreciate the reviewer’s endorsement of our approach to presenting the results of lung tissue analysis. 

  1. Heterogeneous data are reported in the discussion, so no specific conclusion could be reached; in fact, for some metals, the source could not be specified so as to determine potential bioavailability.

Response:  The Discussion summarizes a number of findings from our study, including PM characteristics and elemental composition in lung tissue from a group of well-characterized previously deployed military personnel compared to controls. Our findings support the Conclusion that “the proportion of silica and other silicates was significantly higher in lung tissue from previously deployed military veterans with biopsy-confirmed lung disease compared to controls.”

With regard to the issue of bioavailability, on page 12 second paragraph, we discuss likely exposure sources (anthropogenic vs geogenic), but reflect that the inherent uncertainty with regard to variable sources of exposure limits our ability to determine potential bioavailability.

  1. In addition, confusion related to unknown occupational/environmental exposure still remains an underlying bias.

Response:  We agree that this is a limitation and have acknowledged this several times in the Discussion on page 12 as follows:

Last sentence in the paragraph beginning with ‘Bulk composition . . . ‘:  “Lacking the occupational and environmental exposure histories of control subjects and deployers (beyond deployment), we have little insight into sources of some of the element contents identified.” 

Paragraph on limitations:  “Third, complete occupational and environmental exposure histories for both controls and military deployers were unknown, leading to potential confounding from unknown exposures…Fourth, factors that may lead to increased vulnerability to airborne hazards such as exposure to temperature extremes, stress, noise and acute infections – common in military deployment locations – could not be analyzed”.